# Short Plate with Screw Angle over 20 Degrees Improves the Radiologic Outcome in ACDF: Clinical Study

**DOI:** 10.3390/jcm10092034

**Published:** 2021-05-10

**Authors:** Kathryn-Anne Jimenez, Jihyeon Kim, Jaenam Lee, Hwan-Mo Lee, Seong-Hwan Moon, Kyung-Soo Suk, Hak-Sun Kim, Byung-Ho Lee

**Affiliations:** Orthopedic Department, College of Medicine, Yonsei University, Seoul 03722, Korea; kathrynjimenez@yahoo.com (K.-A.J.); rapport85@yuhs.ac (J.K.); woska0723@yuhs.ac (J.L.); hwanlee@yuhs.ac (H.-M.L.); shmoon@yuhs.ac (S.-H.M.); sks111@yuhs.ac (K.-S.S.); haksunkim@yuhs.ac (H.-S.K.)

**Keywords:** anterior cervical discectomy, plate length, screw insertion angle, subsidence, allospacer

## Abstract

Background: Anterior cervical discectomy and fusion surgery is a common procedure for degenerative cervical spine. This describes allospacer and implant-related outcomes, comparing medium plate–low screw angle and short plate–high screw angle techniques. Methods: From January 2016 to June 2019, 79 patients who underwent ACDF were prospectively enrolled. Patients were divided, depending on the plate–screw system used: medium plate–low screw angle (12.3 ± 2.5 to 13.2 ± 3.2 degrees), and short plate–high screw angle (22.8 ± 5.3 to 23.3 ± 4.7 degrees). Subsidence, ALOD, and sagittal cervical balance were analyzed using lateral cervical X-rays. NDI and VAS scores were also evaluated. Results: Age for medium plate–low-angled screw group is 58.0 ± 11.3 years, and 55.3 ± 12.0 in the short plate–high-angled screw group (*p*-value = 0.313). Groups were comparable in mean NDI (*p*-value = 0.347), VAS (*p*-value = 0.156), C2–C7 SVA, (*p*-value = 0.981), and lordosis angle (*p*-value = 0.836) at 1-year post-surgery. Subsidence was higher in the medium plate–low-angled screw than in the short plate–high-angled screw (25% and 8.5%, respectively, *p*-value = 0.045). ALOD is also more common in the medium plate group (*p*-value = 0.045). Conclusion: Use of a short plate and insertion of high-angled screws (more than 20 degrees) has less chance of subsidence and occurrence of ALOD than the traditional technique of using medium plate and low angle.

## 1. Introduction

Anterior Cervical Discectomy and Fusion (ACDF) is a common surgical procedure performed for degenerative cervical conditions [1]. It has been used for treatment of both myelopathy and radiculopathy with compression mainly on the anterior [2,3].

ACDF is one of the most commonly used surgical treatments for degenerative spine conditions. It is considered a safe procedure, with a low mortality rate. However, it has been observed to have a number of complications such as graft failure, graft subsidence, pseudoarthrosis, adjacent level ossification disease and loss of cervical lordosis [4]. Donor site pain and infection were also reported with the use of autologous bone graft [5].

In a recently published biomechanical study, a finite element model was used to determine the load-sharing ratio between the allospacer and different plate–screw systems. They were able to conclude that a short plate–high angle screw insertion angle provided the best mechanical stability, while medium and maximum length plates have a greater risk of adjacent segment disc disease [6].

There have been numerous plate–screw designs that have been developed to decrease the risk of implant and graft complications. This study investigated allospacer, and implant-related outcomes, comparing the medium plate-low screw angle, and short plate–high screw angle techniques, in patients who underwent ACDF.

## 2. Materials and Methods

This study was approved by the Institutional Review Board of the authors’ hospital (4-2020-1161). From January 2016 to June 2019, a total of 91 patients who underwent ACDF were prospectively enrolled. The patients who had less than 1-year postoperative follow-up, including those with inadequate radiologic images, were excluded from the study. Final patient count included in the study was 79, with 248 segments. Cervical spondylotic myeloradiculopathy (CSMR) is the most common cause of compression with 50 patients (63.3%); 3 patients (3.8%) had ossified posterior longitudinal ligament (OPLL); 26 patients (32.9%) had cervical disc herniation (HCD). There were 47 patients (59.5%) who had short plates–high angle screw (22.8 ± 5.3 to 23.3 ± 4.7 degrees) insertion, and 32 (40.5%) patients with medium plates–low angle screw (12.3 ± 2.5 to 13.2 ± 3.2 degrees) insertion. A total of 19 patients (24.05%) had one-level fusion, 35 (44.3%) had two-level fusion, 20 (25.32%) patients underwent three-level fusion, and 5 (6.33%) patients underwent four-level fusion surgery (Figure 1, flow diagram).

### 2.1. Surgical Indication and Technique

All patients who were included in this study had conservative management prior to being indicated to proceed with surgery. Patients presented with radiculopathy and/or myelopathy.

Surgery was performed via the standard anterior cervical spine approach by one experienced spine surgeon. Cervical discectomy and adequate decompression were completed with the use of a microscope. The disc space was prepared, preserving as much end plate as possible. The size of the allospacer (CORNERSTONE^TM^ ASR, Medtronic Sofamor Danek, Inc., Memphis, TN, USA) was chosen carefully, with C-arm guidance to prevent over distraction of the segment. The allospacer was placed in the ventral position, with contact on the anterior cortex of the body. There were two available plate and screw systems used. In the early part of this study, the medium plate with low-angled screws was used (ATLANTIS VISION^®^ Elite Anterior Cervical Plate System, Medtronic Sofamor Danek, Inc., Memphis, TN, USA), and the short plate–high-angled screw (ZEVO^TM^ ANTERIOR CERVICAL PLATE SYSTEM ASR, Medtronic Sofamor Danek, Inc., Memphis, TN, USA) was used in the latter part of this study. The ATLANTIS VISION plate is a combination of a fixed and variable angle screw system that allows a screw angle of 10–15 degrees. The ZEVO plate is a variable angle screw system that allows the placement of a higher-angled screw (≥20 degrees). Intra-operative images were taken to check the correct placement of the allospacer and the implant [7]. The same post-operative treatment protocol was used in all the patients.

### 2.2. Evaluation of Radiologic Outcomes

The lateral cervical spine X-rays were taken pre-operatively, immediately post-surgery, at 3-monthsfollow-up, and at 1-year follow-up. The C2–7 cervical lordosis (CL) and C2–7 sagittal vertical axis were used for the cervical sagittal balance. Improvement in cervical sagittal balance has been proven to be important in prevention of adjacent segment disease requiring revision surgery and has shown to have a better clinical outcome [8,9,10].

The same post-operative X-rays were used to measure the plate-to-disc distance, superior and inferior screw angles, and presence of subsidence. The presence of screw pull-out, allospacer breakage, or dislodgement were also recorded. Subsidence is more than a 3 mm decrease in height when compared to the immediate postoperative period [11].

The measurements taken are defined as follows:C2–7 Cervical Lordosis (CL): Cobb’s Method. This was measured by the angle formed by the perpendicular lines parallel to the inferior endplates of C2 and C7 [12].C2–7 Sagittal Vertical Axis (SVA): the distance between the C2 plumb line and the posterosuperior corner of C7 [12].Plate-to-Disc Distance: distance from the most proximal part and most distal part of the plate to the superior and inferior adjacent disc spaces, respectively [13].Screw Angles: angle between a line perpendicular to the plate and the direction of the screw [6].Subsidence: the vertical length from the superior endplate of the most superior vertebra to the inferior endplate of the most inferior vertebra [14].

For the clinical outcome of the patients, the Neck Disability Index (NDI) and Euro-Qol-Visual Analogue Scale (VAS) scores were noted during the preoperative period, on the 3-month and 1-year post-operative follow-ups [15,16].

### 2.3. Statistical Analysis

Independent *t*-test and Chi-square test were used to evaluate the differences between the two groups. Bonferroni adjustment was carried out to confirm the statistical differences. Linear regression was also analyzed to determine the relationship of the variables between the two groups. *p* values less than 0.05 were considered statistically significant.

## 3. Results

In terms of patient characteristics, the two groups only differed significantly in terms of sex (*p*-value = 0.035), diagnosis (*p*-value = 0.003), and plate-to-disc distance (*p*-value < 0.001). Anatomically, females tend to have smaller cervical vertebra as compared to males. This difference in anatomy allows the male patients to accommodate a shorter plate, with a bigger plate-to-disc distance. In terms of diagnosis, patients who had the short plate were diagnosed with CSMR (48.9%), OPLL (6.4%), and HCD (44.7%). Among the patients who had the medium plate, none had OPLL, while 84.4% were diagnosed with CSMR and 15.6% with HCD. In the short plate group, the superior plate-to-disc distance was 10.7 ± 2.3 and the inferior plate-to-disc distance was 9.0 ± 2.0. In the medium plate group, the superior plate-to-disc distance was 4.7 ± 1.8, and the inferior plate-to-disc distance was 4.0 ± 1.0. (Table 1)

The two groups were comparable in terms of mean NDI during pre-operative period (*p*-value = 0.060; marginally significant), 3 months post-surgery (*p*-value = 0.445) and 1 year post-surgery (*p*-value = 0.465). Additionally, they were comparable in mean VAS scores during pre-operative period (*p*-value = 0.993), 3 months post-surgery (*p*-value = 0.158) and 1 year post-surgery (*p*-value = 0.124).

### 3.1. Cervical Spine Sagittal Balance

The two groups did not differ significantly in C2–C7 SVA during the pre-operative period (*p*-value = 0.918), 3 months post-surgery (*p*-value = 0.635) and 1 year post-surgery (*p*-value = 0.510).

The mean C2–C7 cervical lordosis angle was higher among patients with short plates compared to those with medium plates during the pre-operative period (*p*-value = 0.627), 3 months post-surgery (*p*-value = 0.394) and 1 year post-surgery (*p*-value = 0.600), but values were not statistically significant. Likewise, the proportion of patients with cervical lordosis angle <20 was comparable during pre-operative period (*p*-value = 0.607), 3 months post-surgery (*p*-value = 0.551) and 1 year post-surgery (*p*-value = 0.524).

### 3.2. Subsidence and Adjacent Level Ossified Disease (ALOD)

The two groups differed significantly in superior and inferior screw angles during immediate post-operation 3 months after, and 1 year after (*p*-values < 0.001). The mean subsidence rate was 1.9 ± 1.8 mm in the short plate group and 3.2 ± 2.7 mm in the medium plate group at 1 year post-surgery. The proportion of patients with >3 mm subsidence was also significantly higher among those with medium plate compared to those with short plate (25.0% versus 8.5%, *p*-value = 0.045). The presence of ALOD was also significantly higher in the medium plate group than the short plate group (31.2% versus 12.8%, *p*-value = 0.045) (Table 2).

After adjusting for sex and diagnosis, significant difference in patient outcomes were observed between the two groups in terms of subsidence rate. Those with medium plates significantly had higher subsidence rates compared with those with short plates at 3 months post-surgery (regression coefficient = 1.3, 95% CI = 0.1 to 2.4, *p*-value = 0.033) and at the 1-year follow-up (regression coefficient = 1.4, 95% CI -0.2 to 2.5, *p*-value = 0.018). It is important to note that after adjustments, the *p*-value was significantly lower as compared to before analyzing with adjustments at 3 months post-surgery (*p*-value = 0.133 before adjustment) (Table 3, Figure 2 and Figure 3).

## 4. Discussion

ACDF is regarded as one of the most widely utilized surgeries for cervical degenerative spine condition. The main goal of this procedure is decompression of the affected segments, as well as restoring spinal alignment and providing stability [2,3,17]. Implant-related complications, such as graft failure, subsidence and screw pullout are major concerns in ACDF-only surgeries. Graft collapse has been reported to occur at 28.8% [18,19]. The risk of subsidence with stand-alone cage or allospacer still remain as one of the main complications with this technique. However, it has been proven that, with proper technique and implant choice, this risk can be greatly reduced [5,11,20]. The addition of plating in ACDF, even for single level, has led to earlier fusion and improved clinical outcomes in longer fusion [21].

Subsidence has been defined as the sinking of an object (allospacer) with a higher modulus of elasticity to an object with a lower modulus of elasticity (vertebral body) [22]. A small amount of subsidence may help in achieving bone fusion and may be part of the integration process between the implant and the bone [23]. Excessive subsidence, however, could cause loss of cervical lordosis leading to kyphotic deformity, screw pull-out, and breakage [20,22]. It is important to limit subsidence by attaining a stable construct during surgery. In a study by Kwon et al., their subsidence risk analysis showed that short plates were able to achieve a high insertion screw angle due to the geometry of the cervical spine. The study was able to demonstrate that a short plate–high-angled screw construct reduces the stress on the endplate, which, in turn, reduces the risk of subsidence [6]. It was also noted that the use of a long screw increases the pullout strength, thus improving the stability of the implant construct [24]. The placement of the allospacer in the intervertebral space is also crucial in decreasing the risk of subsidence. In another biomechanical study, it was observed that the anterior placement of the allospacer, which provides an anterior cortical bone bite, reduces subsidence [7]. The mean incidence of subsidence is at 21.1% [3]. This study showed similar results with the medium plate–low-angled screws group at 25%. The drastic difference with the 8.5% incidence in short plate–high-angled screws confirms the finite element analysis of Kwon et al. [6]. This study also showed higher insertion screw angle of more than 20 degrees as compared to other studies that used a variable plate screw system with an average of only 12 degree screw insertion angle, based on the screw design which permits a maximum angle of 12–15 degrees [25,26].

Adjacent level ossification disease has been seen in 45–67% of the cases of patients who underwent ACDF with a plate-to-disc distance of less than 5 mm [27]. This may be attributed to the need for a wider dissection needed to place a longer plate [27,28]. Our study showed an incidence of 12.8% of ALOD in the short plate–high-angled screw group, which is a considerable difference compared to the 31.2% incidence in the medium plate group. Another advantage in using a short plate when the need arises for revision surgery of an adjacent segment is that the removal of the previously inserted plates and screws is not necessary. This would minimize excessive dissection of the area [6,29].

Restoration of cervical sagittal balance is important in the clinical outcome of patients undergoing any cervical spine procedure. An imbalance in the alignment in this region has been associated with an increased risk in adjacent segment pathology, which could lead to revision surgery [7,8]. In this study, the two groups were comparable in their sagittal balance. The short plate–high-angled screw group was able to achieve a higher lordosis angle, but it was not statistically significant.

## 5. Conclusions

In patients undergoing ACDF, the use of a short plate and the insertion of high-angled screws (more than 20 degrees) have less chance of subsidence and occurrence of ALOD than the traditional technique of using a medium plate and low-angled screws.

## Figures and Tables

**Figure 1 jcm-10-02034-f001:**
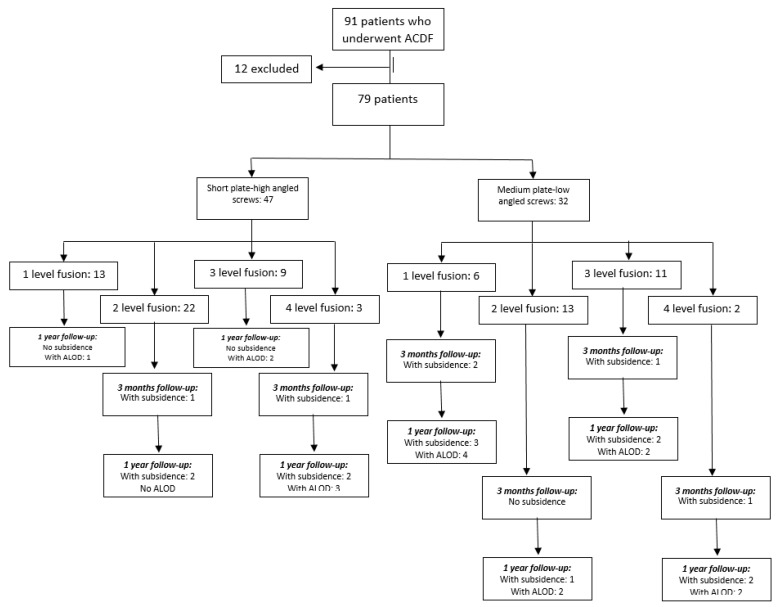
Flow diagram of methodology and results.

**Figure 2 jcm-10-02034-f002:**
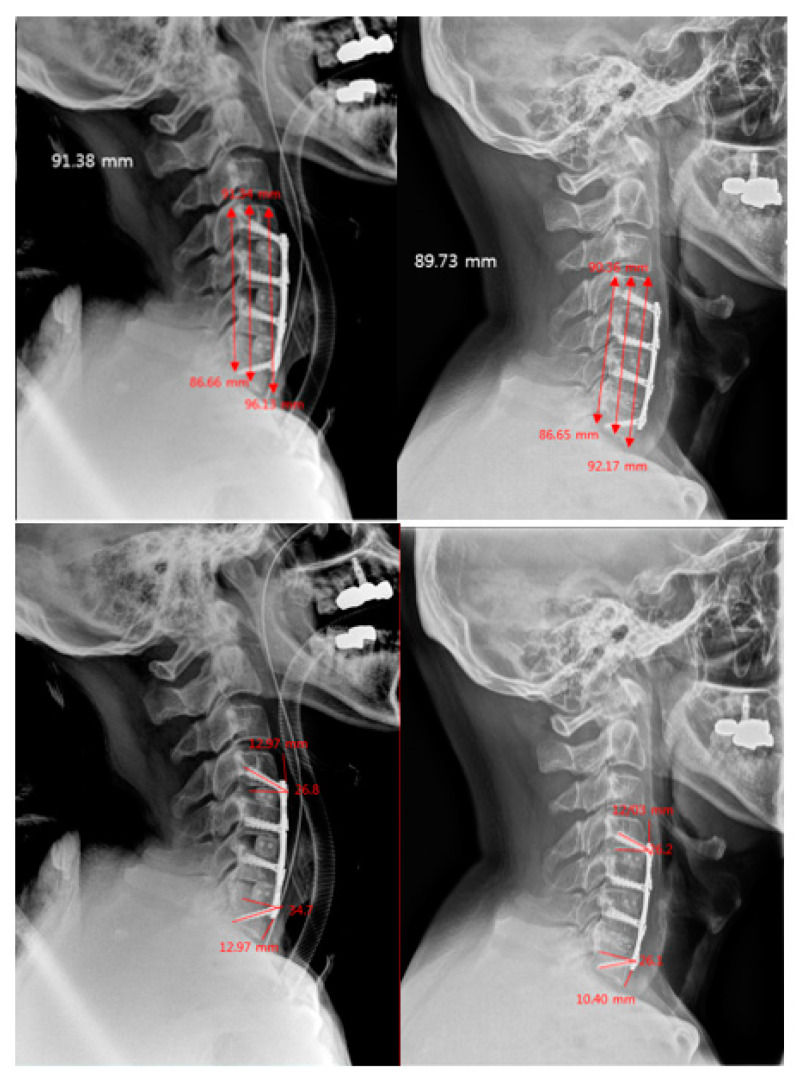
Short plate–high-angled screw technique, 3 level fusion. Images on the left show the immediate post-operative X-rays; the right show the X-rays performed at the 1-year follow-up. Comparing the images, it shows minimal subsidence and change in PPD and screw angles at 1 year.

**Figure 3 jcm-10-02034-f003:**
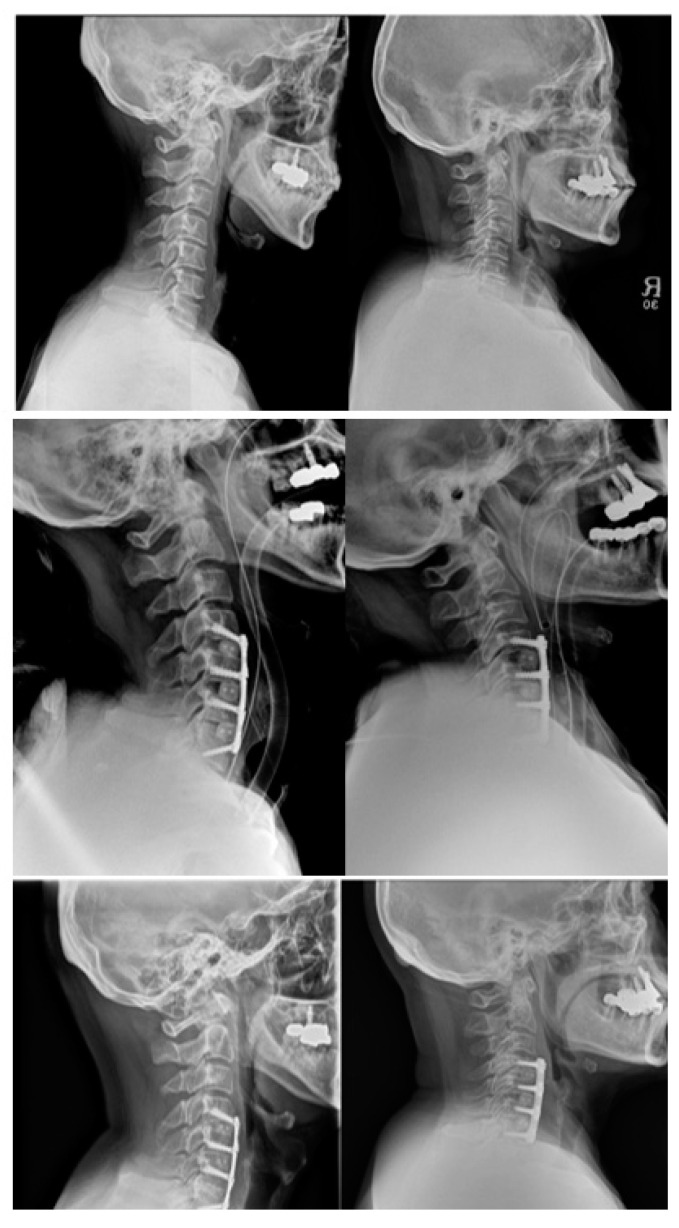
Side by side comparison of the two techniques (images on the left, short plate–high-angled screws; images on the right, medium plate–low-angled screws) at pre-op, immediate post-op and at the 1-year follow-up. Images show presence of subsidence on the medium plate–low-angled screw, with decrease in PPD. The lower adjacent segment at 1-year post-op also showed the presence of ALOD.

**Table 1 jcm-10-02034-t001:** Comparison of demographic characteristics between the 2 groups.

	Short Plate (*n* = 47)	Medium Plate (*n* = 32)	Overall	*p*-Value
Sex				0.035
Male	29 (61.7)	12 (37.5)	41 (51.9)	
Female	18 (38.3)	20 (62.5)	38 (48.1)	
Age	55.3 ± 12.0	58.0 ± 11.3	56.4 ± 11.7	0.313
Diagnosis				0.003
CSMR	23 (48.9)	27 (84.4)	50 (63.3)	
OPLL	3 (6.4)	0 (0.0)	3 (3.8)	
HCD	21 (44.7)	5 (15.6)	26 (32.9)	
Myelopathy	33 (70.2)	24 (75.0)	57 (72.2)	0.641
Operation Level				0.476
1	13 (27.7)	6 (18.8)	19 (24.1)	
2	22 (46.8)	13 (40.6)	35 (44.3)	
3	9 (19.2)	11 (34.4)	20 (25.3)	
4	3 (6.4)	2 (6.3)	5 (6.3)	
Plate-to-Disc Distance (mm) Superior	10.7 ± 2.3	4.7 ± 1.8	8.3 ± 3.6	<0.001
Plate-to-Disc Distance (mm) Inferior	9.0 ± 2.0	4.0 ± 1.0	7.0 ± 3.0	<0.001

Note: CSMR: Cervical Spondylotic Myeloradiculopathy, OPLL: Ossified Posterior Longitudinal Ligament, HCD: Herniated Cervical Disc.

**Table 2 jcm-10-02034-t002:** Subsidence and presence of ALOD between short and medium plate.

	Short Plate (*n* = 47)	Medium Plate (*n* = 32)	Overall	*p*-Value
Immediate Post-op				
Screw Angle Superior	22.8 ± 5.3	12.3 ± 2.5	18.5 ± 6.8	<0.001
Screw Angle Inferior	23.3 ± 4.7	13.2 ± 3.2	19.2 ± 6.5	<0.001
Control for Subsidence	61.6 ± 19.8	57.3 ± 22.5	59.9 ± 20.9	0.375
3 Months				
Screw Angle Superior	21.4 ± 5.0	11.0 ± 2.5	17.2 ± 6.6	<0.001
Screw Angle Inferior	22.0 ± 4.6	11.8 ± 3.6	17.9 ± 6.5	<0.001
Subsidence Rate	2.1 ± 1.9	2.9 ± 2.8	2.4 ± 2.3	0.133
Subsidence >3 mm (n, %)	2 (4.26)	4 (12.5)	6 (7.59)	0.179
1 Year				
Screw Angle Superior	20.9 ± 5.0	10.7 ± 2.6	16.7 ± 6.6	<0.001
Screw Angle Inferior	20.9 ± 5.0	11.2 ± 3.5	17.0 ± 6.5	<0.001
Subsidence Rate	1.9 ± 1.8	3.2 ± 2.7	2.5 ± 2.3	0.017
Subsidence >3 mm (n, %)	4 (8.5)	8 (25.0)	12 (15.2)	0.045
ALOD	6 (12.8)	10 (31.2)	16 (20.2)	0.045

Note: ALOD: Adjacent Level Ossified Disease.

**Table 3 jcm-10-02034-t003:** Regression analysis on post-operative outcomes between short and medium plate to adjust for sex and diagnosis.

	Regression Coefficient	95% Confidence Interval	*p*-Value
3 months			
Subsidence Rate	1.3	0.1 to 2.4	0.033
C2–C7 SVA	−0.7	−4.3 to 3.0	0.716
Cervical Lordosis Angle	−1.8	−6.0 to 2.4	0.388
NDI	−0.05	−0.1 to 0.02	0.162
VAS	6.4	−0.9 to 13.8	0.087
1 year			
Subsidence Rate	1.4	0.2 to 2.5	0.018
C2–C7 SVA	0.04	−3.2 to 3.3	0.981
Cervical Lordosis Angle	0.3	−3.0 to 3.8	0.836
NDI	−0.02	−0.1 to 0.03	0.347
VAS	4.8	−1.9 to 11.4	0.156

Note: SVA: Sagittal Vertical Axis, NDI: Neck Disability Index, VAS: Visual Analogue Scale.

## Data Availability

Not applicable.

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
