# Peer review of "Short Plate with Screw Angle over 20 Degrees Improves the Radiologic Outcome in ACDF: Clinical Study"

_jcm, 2021, doi:10.3390/jcm10092034_

Round 1

Reviewer 1 Report

Authors present a prospective clinical study on 79 patients who underwent anterior cervical discectomy and fusion, where allospacer and implant related outcomes, comparing medium plate-low screw angle and short plate-high screw angle technique were compared and analyzed. In accordance to previously published biomechanical studies, the authors found that use of a short plate and insertion of high angled screws (more than 20 degrees) has less chance of subsidence and occurrence of adjacent segment disease than the traditional technique of using medium plate and low angle screws. One limitation of this study is that the authors treated patients with several different pathologies; it would be interesting to perform a separate statistic analysis of patients who harbored only cervical spinal canal stenosis with myelopathy to check if there were any differences among one group of patients. Also several illustrative cases with MRI and CTs pre- and postoperative could add additional value to this clinical study. Taking the ongoing debate on To plate vs. Not to plate in the anterior cervical spine surgery it would be interesting to include this point into the discussion section and to give it a thought if subsidence would be less present if there was not any plate at all. Discussion could also be enriched on a literature review with the most important studies which discussed the question of a plate design and its influence on subsidence and fusion.

Reviewer 2 Report

Without explaining why why for the statistical analysis after applying
the T and Chi test of the Bonferoni correction - which is dedicated
to multiple comparisons, and why linear regression was used
not the correlation.Please explain.

Round 2

Reviewer 2 Report

The article, after taking into account the reviewers 'corrections and the authors' reference to statistical doubts, is ready for publication in its present form.